# Neuroprotective and Anti-Inflammatory Effects of Dimethyl Fumarate, Monomethyl Fumarate, and Cannabidiol in Neurons and Microglia

**DOI:** 10.3390/ijms252313082

**Published:** 2024-12-05

**Authors:** Alicia Sánchez-Sanz, María José Coronado-Albi, Rafael Muñoz-Viana, Antonio García-Merino, Antonio J. Sánchez-López

**Affiliations:** 1Neuroimmunology Unit, Instituto de Investigación Sanitaria Puerta de Hierro-Segovia de Arana, 28222 Madrid, Spain; 2Confocal Microscopy Core Facility, Instituto de Investigación Sanitaria Puerta de Hierro-Segovia de Arana, 28222 Madrid, Spain; 3Bioinformatics Unit, Instituto de Investigación Sanitaria Puerta de Hierro-Segovia de Arana, 28222 Madrid, Spain; 4Department of Neurology, Hospital Universitario Puerta de Hierro Majadahonda, 28222 Madrid, Spain; 5Department of Medicine, Universidad Autónoma de Madrid, 28049 Madrid, Spain; 6Red Española de Esclerosis Múltiple (REEM), 08028 Barcelona, Spain; 7Biobank, Instituto de Investigación Sanitaria Puerta de Hierro-Segovia de Arana, 28222 Madrid, Spain

**Keywords:** dimethyl fumarate, monomethyl fumarate, cannabidiol, multiple sclerosis, neuroprotection, microglia, neuron, transcriptome, Nrf2, NF-kB

## Abstract

Dimethyl fumarate (DMF) is an immunomodulatory treatment for multiple sclerosis (MS) that can cross the blood–brain barrier, presenting neuroprotective potential. Its mechanism of action is not fully understood, and there is a need to characterize whether DMF or its bioactive metabolite monomethyl fumarate (MMF) exerts neuroprotective properties. Moreover, the combination of adjuvant agents such as cannabidiol (CBD) could be relevant to enhance neuroprotection. The aim of this study was to compare the neuroprotective and immunomodulatory effects of DMF, MMF, and CBD in neurons and microglia in vitro. We found that DMF and CBD, but not MMF, activated the Nrf2 antioxidant pathway in neurons. Similarly, only DMF and CBD, but not MMF, prevented the LPS-induced activation of the inflammatory pathway NF-kB in microglia. Additionally, the three drugs inhibited the production of nitric oxide in microglia and protected neurons against apoptosis. Transcriptomically, DMF modulated a greater number of inflammatory and Nrf2-related genes compared to MMF and CBD in both neurons and microglia. Our results show that DMF and MMF, despite being structurally related, present differences in their mechanisms of action that could be relevant for the achievement of neuroprotection in MS patients. Additionally, CBD shows potential as a neuroprotective agent.

## 1. Introduction

Dimethyl fumarate (DMF) is a diester of the simple organic acid fumaric acid, which is found in the plant *Fumaria officinalis*, from which it obtains its name [1]. DMF was first approved in Germany in 1994 for the treatment of the autoimmune disease psoriasis, under the name Fumaderm^®^ (Biogen, Cambridge, MA, USA), which is composed of a mixture of ≈60% DMF and other fumaric acid esters [2]. In 2014, DMF was approved in Europe as an immunomodulator-based monotherapy for relapsing-remitting multiple sclerosis (RRMS) (Tecfidera^®^, Biogen, Cambridge, MA, USA; also known as BG-12) [3]. Multiple sclerosis (MS) is an autoimmune disease that affects the myelin sheath of the central nervous system (CNS) axons, leading to axonal damage and neurodegeneration. The pathophysiology of MS is complex and includes the infiltration into the CNS through a disrupted blood–brain barrier (BBB) of peripheral immune cells comprising self-reactive lymphocytes and myeloid cells. These cells, together with the CNS-resident glial cells microglia and astrocytes, lead to inflammation, demyelination, neurodegeneration, and gliosis [4].

All MS therapies approved to date target the peripheral immune system, achieving a reduction in the number of relapses that characterize the inflammatory phase of the disease. However, there is an unmet need for halting the progression of neurodegeneration, and currently, there are no approved therapies with demonstrated neuroprotective or remyelinating properties [5]. Moreover, the available therapies have limited efficacy in the control of the progressive forms of MS, highlighting the role of compartmentalized CNS inflammation in driving disease progression independent of relapse activity [6]. Most of the authorized therapies do not act beyond the BBB, and thus are unable to modulate glial responses [7]. DMF could be relevant in this respect as it is a small molecule that can cross the BBB and could exert a beneficial dual effect inside the CNS in addition to the immunomodulation of the peripheral immune system, which confers it with the potential to control the progression of neurodegeneration beyond its demonstrated efficacy in clinical trials in the prevention of relapses [8,9].

DMF has exhibited neuroprotective effects in in vitro and in vivo murine models of MS, although its impact on the CNS of MS patients remains unknown. DMF protects neurons and oligodendrocytes against oxidative stress and apoptosis in vitro and preserves myelin and axons in the experimental autoimmune encephalomyelitis (EAE) model [10,11,12]. In addition, DMF has also shown anti-inflammatory effects in microglia and astrocytes, resulting in an indirect neuroprotective effect through the inhibition of the production of neurotoxic inflammatory mediators [13,14]. However, most studies have focused on the direct effects of DMF on neurons and glial cells, despite the unlikeliness of an in vivo exposure of cells to this molecule. DMF is an oral drug that, upon intake, is mostly metabolized by esterases in the intestine, resulting in its hydrolyzed form monomethyl fumarate (MMF), which is the main metabolite detected in plasma and is considered the biologically active compound [15]. In addition, MMF is the only metabolite that has been detected in the cerebrospinal fluid of MS patients [16].

The molecular mechanism of action of DMF is not fully understood and is considered multifactorial. DMF can react with thiol groups present in cysteine residues in a reaction known as succination, which can modify the functionality of the targeted proteins, interfering in diverse signaling pathways [1]. Among the identified targets of DMF are the nuclear factor erythroid 2-related factor 2 (Nrf2) antioxidant pathway, the nuclear factor kappa B (NF-kB) inflammatory pathway, the hydroxycarboxylic acid receptor 2 (HCAR2), the glycolytic enzyme GAPDH, or the antioxidant peptide glutathione (GSH) [12,13,17,18,19]. Most of the studies elucidating the mechanism of action of fumarates have focused on DMF, treating DMF and MMF interchangeably without considering their metabolism and pharmacology. When exposed to organ preparations including pancreatic extract, intestinal homogenate and perfusate, or liver fraction, DMF is rapidly hydrolyzed within minutes, while MMF shows a much longer degradation half-life. In addition, it has been shown that DMF penetrates CACO-2 cell monolayers much faster than MMF [20]. In line with this, it was shown that DMF reacts quickly with GSH inside cells, and GSH-conjugates with DMF have been detected in urine and blood, while MMF is less reactive toward GSH [21,22,23]. Moreover, there is evidence of a differential sensitivity of cysteines to both fumarates. One study found that several proteins related to T-cell function including protein kinase Cθ presented cysteines that were sensitive to DMF, but not to MMF, resulting in an inhibition of T cells achieved by DMF, but not by MMF [24]. In addition, distinct effects between DMF and MMF have been described on the NF-kB pathway where only DMF, but not other fumarates, inhibited NF-κB activity, and on the modulation of glial cells, where only DMF, but not MMF, presented significant anti-inflammatory and antioxidant effects in microglia and astrocytes [25,26,27]. Despite this, there is a lack of studies comparing the effects of both fumarates, which could be relevant to an understanding of their real in vivo effects and for the achievement of neuroprotection in MS patients.

In addition, the use of adjuvant therapies could be a suitable strategy to achieve neuroprotection in MS. Cannabidiol (CBD) is a cannabinoid present in the plant *Cannabis sativa* that has therapeutic potential in MS due to its antioxidant and anti-inflammatory properties [28]. CBD is already approved as a symptomatic treatment in MS as nabiximol, which is composed of a mixture of 1:1 of Δ-9-tetrahidrocannabinol (THC) and CBD and presents a positive tolerability profile. Moreover, CBD has advantages over THC due to its lack of psychotropic effects [29]. In murine models of MS, CBD ameliorates clinical symptoms and inhibits microglia activation and the production of inflammatory cytokines [30,31]. In addition, CBD has shown direct neuroprotective properties in EAE, protecting oligodendrocytes against oxidative stress and diminishing axonal damage and demyelination [32,33]. The mechanism of action of CBD is not fully understood, but it has been described that CBD activates Nrf2 and depletes GSH, effects also observed in vitro with DMF [12,13,34]. In addition, DMF has also been related to the endocannabinoid system through the modulation of the expression of the cannabinoid type 2 (CB2) receptor in microglia, which is also activated by CBD, and through the modulation of the levels of endocannabinoids in MS patients [35,36]. The anti-inflammatory and antioxidant properties of CBD, together with the shared pathways with DMF, suggests the potential use of CBD as an adjuvant therapy to DMF in neuroprotection.

The aim of the present study was to compare the molecular mechanisms of action related to the neuroprotection of DMF and MMF and to evaluate the potential use of CBD as an adjuvant therapy.

## 2. Results

### 2.1. Effect of DMF, MMF, and CBD on the Cell Viability of Neurons and Microglia

Prior to the experiments, the potential toxicity of the three drugs (DMF, MMF, and CBD) was tested in neurons and microglia to select the adequate working concentrations (Figure 1). Cell viability was measured at 24, 48, and 72 h of treatment with different drug concentration ranges. For each cellular type, we determined the half-maximal inhibitory concentration (IC_50_) of each drug. In neurons, as a non-proliferative primary culture, the IC_50_ was defined as the drug concentration able to diminish cell viability by 50% with respect to the vehicle-treated control. In microglia, as a proliferative cell line, the IC_50_ was defined as the drug concentration able of inhibiting by 50% cellular proliferation with respect to the vehicle-treated control. The initial dose range tested for DMF and MMF was chosen according to the concentration of MMF detected in the brain (approximately 100 µM) after oral intake of the therapeutic dose established for DMF [9]. For CBD, the dose range was chosen according to the concentration (approximately 10 µM) that has demonstrated antioxidant and anti-inflammatory effects in vitro [37].

The IC_50_ of DMF in neurons was around 100 µM at 24 h, while at 72 h, a much lower dose of DMF (36 µM) already produced 50% of cell death (Figure 1A). In microglia, the IC_50_ of DMF at 24 h was 37 µM, but a higher dose (73 µM) was needed at 72 h to inhibit cell proliferation by 50% (Figure 1B). For MMF, the IC_50_ in neurons at 24 h was around 65 µM, indicating a higher toxicity of MMF compared to the IC_50_ of DMF at the same time point (Figure 1C). In addition, and in contrast with DMF, which showed a higher toxicity over time, the IC_50_ of MMF at 72 h remained almost the same (55 µM) as at 24 h. In microglia, the IC_50_ of MMF was almost constant at around 50 µM at all of the studied timepoints (Figure 1D). Taken together, these results indicate that DMF and MMF present different pharmacodynamics compared to each other, and they also exhibit different pharmacodynamics depending on the cellular subtype. Regarding CBD, the IC_50_ in neurons was 15 µM at 24 h and decreased slightly to 13 µM at 72 h (Figure 1E). In microglia, the IC_50_ of CBD was constant at around 12 µM at all timepoints (Figure 1F). Based on these findings, we selected 30 µM for DMF and MMF and 10 µM for CBD as the maximum experimental concentrations to work at doses under the IC_50_.

### 2.2. Differential Effects of DMF, MMF, and CBD on the Activation of Nrf2 in Neurons and Microglia

Activation of the Nrf2 transcription factor has been proposed as one of the main mechanisms of action of DMF, with this pathway being of special relevance in neuroprotection as it promotes the upregulation of antioxidant target genes, conferring cells with resistance to oxidative damage [12]. However, some studies have described discrepancies in the ability of DMF and MMF to activate the Nrf2 pathway, suggesting a lack or lower Nrf2 activation by MMF compared to DMF [27,38]. Thus, we aimed to compare the ability of DMF, MMF, and CBD to activate Nrf2 at the protein level in neurons and microglia (Figure 2). In neurons, DMF induced the translocation of Nrf2 to the nucleus at 30 µM compared to the vehicle (*p* = 0.0275), but not at lower concentrations (Figure 2A). In microglia, DMF also induced the nuclear translocation of Nrf2, starting from a lower concentration (10 µM) (*p* < 0.0001) than in neurons (Figure 2A). This activation in microglia was also maintained at 30 µM DMF (*p* = 0.0070). Furthermore, the DMF-induced activation of Nrf2 was of a higher magnitude in microglia than in neurons (10-fold in microglia compared to 4-fold in neurons), indicating that DMF exerts a greater Nrf2 activation in microglia than in neurons. As for MMF, we did not observe the nuclear translocation of Nrf2 in neurons at the same concentrations of DMF (Figure 2B). Similarly, we did not observe Nrf2 activation by MMF in the microglial cells at any concentration (Figure 2B). These results suggest that DMF and MMF exert differential effects on the activation of the Nrf2 pathway.

Regarding CBD, we observed the activation of Nrf2 starting at the dose of 5 µM (*p* = 0.0004), which induced a 4-fold increase in the Nrf2 nuclear protein levels compared to the vehicle-treated cells (Figure 2C). In addition, Nrf2 activation increased to a 10-fold magnitude compared to the vehicle at 6 and 7 µM (*p* < 0.0001 for both) and then decreased again to a 4-fold change at higher CBD concentrations (8 and 10 µM, *p* = 0.0004 and *p* = 0.0048, respectively), indicating that there is a peak activation of Nrf2 induced by CBD. Importantly, the peak of Nrf2 activation induced by CBD in neurons was higher than that induced by DMF (10-fold by CBD compared to 4-fold by DMF), highlighting the potential of CBD as an antioxidant adjuvant therapy. Interestingly, in microglial cells, CBD did not produce the activation of Nrf2, indicating that CBD exerts differential effects in neurons and microglia (Figure 2C). Representative confocal microscopy images of the Nrf2 protein in neurons and in microglia are shown in Figure 2D,E, respectively.

### 2.3. Immunomodulatory Effects of DMF, MMF, and CBD in Microglia

To evaluate the anti-inflammatory properties of DMF, MMF, and CBD in microglia, we studied their ability to inhibit the NF-kB pathway and the production of nitric oxide in the LPS-activated cells. From the family of inflammatory NF-kB transcription factors, we studied the inhibition of the p65 subunit (RelA), which has been described to be covalently modified by DMF [39]. First, we verified that LPS treatment induced the nuclear translocation of NF-kB p65 (Figure 3A–C). DMF treatment completely prevented NF-kB p65 translocation at 10 and 30 µM (*p* = 0.0048 and *p* < 0.0001, respectively), observing the same nuclear protein levels as in the vehicle-treated cells (Figure 3A). In contrast, MMF did not prevent the activation of NF-KB at any of the studied concentrations (Figure 3B), indicating that DMF and MMF exert differential effects on the NF-kB pathway in microglia. Regarding CBD, we could observe the inhibition of NF-kB p65 at 4 and 6 µM (*p* = 0.0352 and *p* = 0.0183, respectively), although this inhibition was not achieved at higher CBD concentrations (Figure 3C). Thus, both CBD and DMF, but not MMF, are able to inhibit the NF-kb p65 pathway in microglia. However, the inhibition of NF-kB p65 by CBD seemed less effective than with DMF, as we could still observe some level of the nuclear translocation of NF-kB p65 in the CBD-treated cells, while in the DMF-treated cells, this translocation was completely abolished. Representative confocal microscopy images of the NF-kB p65 protein in microglia are shown in Appendix A.

In addition, we studied the concentration of NO in the supernatant of the microglial cell cultures as an indicator of an activated pro-inflammatory state. LPS treatment triggered the release of NO compared to the vehicle at the three studied timepoints, detecting the highest NO concentrations (around 40 µM) at 48 and 72 h (Figure 3D–F). DMF significantly inhibited the production of NO at 48 and 72 h, but not at 24 h (Figure 3D). This inhibition was observed at all of the studied DMF concentrations, even at the lowest dose of 1 µM, achieving a reduction by half of the concentration of NO (around 20 µM) compared to the LPS-treated cells. MMF produced a similar inhibition at the same concentrations as DMF, halving the concentrations of NO compared to the LPS-treated cells (Figure 3E). However, in contrast with DMF, MMF showed a significant NO inhibition from the earliest studied timepoint (24 h). CBD also inhibited the production of NO at all timepoints, but this inhibition was only achieved starting from 6 µM CBD and not at lower CBD concentrations (Figure 3F). Moreover, CBD showed a greater inhibition than DMF and MMF as the NO concentrations detected in the CBD-treated cells were below 10 µM, and the concentration of NO was almost undetectable in the cells treated with 10 µM CBD.

Finally, we aimed to directly study the neuroprotective effects of DMF, MMF, and CBD by comparing the ability of the three drugs to protect neurons from apoptosis. Given that physiologically, if any of the drugs enter the BBB, both the neurons and microglia will be exposed to them, we decided to study the synergistic effects (direct antioxidant properties in neurons with indirect anti-inflammatory effects in microglia) of the drugs in both cells. With this aim, neurons were pre-treated with DMF, MMF, or CBD for 4 h and then treated for another 4 h with the supernatant from the activated microglial cells treated with LPS alone or combined with DMF, MMF, or CBD at the same concentrations as in the neurons. First, we assessed whether the conditioned media from the LPS-treated microglial cells significantly increased neuronal apoptosis (Figure 3G–I). DMF treatment significantly prevented neuronal apoptosis at 10 µM (*p* = 0.0032), but not at 30 µM (Figure 3G). MMF also prevented neuronal apoptosis at 10 µM (*p* = 0.0007), but in contrast with DMF, MMF at 30 µM also protected neurons from apoptosis (*p* = 0.0003) (Figure 3H). CBD prevented neuronal apoptosis at 7 µM (*p* = 0.0190) (Figure 3I). However, although we observed a reduction in neuronal apoptosis in the CBD-treated cells compared to LPS alone, this reduction was to a lesser extent than with DMF or MMF, in which neuronal apoptosis was comparable to the vehicle-treated cells. Representative confocal microscopy images of the apoptosis assay are shown in Appendix A.

### 2.4. Transcriptomic Effects of DMF, MMF, and CBD in Neurons

To deepen our knowledge of the molecular mechanism of action of DMF, MMF, and CBD, we carried out a transcriptomic study of neurons. Figure 4A shows the principal component analysis (PCA) of neurons treated with DMF, MMF, or CBD and their respective vehicles for 4 or 24 h. Most of the samples were grouped in the same region of space, with no clear differentiation between the drugs and vehicles, indicating that DMF, MMF, and CBD exerted a mild transcriptional effect. In addition, the aggrupation of the three drugs in the same region of space indicates that they could exert similar transcriptomic changes in agreement with the activation of similar pathways. Only the samples treated with DMF for 4 h were located far away from the rest of the samples in the PCA and separated noticeably from their respective vehicle, indicating that this treatment triggered a considerable change in neuronal transcription (Figure 4A and Appendix A). However, this spatial clustering was not maintained in the 24 h DMF-treated samples, which were located in the PCA closer to the samples treated only with the vehicle (DMSO), indicating that the strong transcriptomic alteration caused by DMF is lost over time. Moreover, we found no differential clustering for the rest of samples depending on the time of treatment. Samples treated for 4 h or 24 h tended to be located in nearby regions of the PCA space. This observation was supported by the near absence of differentially expressed genes (DEGs) between 24 h and 4 h for each drug treatment condition, except for DMF, which presented 1276 DEGs across time (Appendix A).

Regarding the number of DEGs induced by each drug, DMF modulated a total number of 407 DEGs compared to the vehicle after 4 h of treatment (Figure 4B and Appendix A). From these DEGs, 169 were downregulated and 238 were upregulated (Appendix A). In addition, 169 DEGs presented strong fold changes (>1 or <−1), and most of the genes with strong fold changes (131 DEGs) were upregulated (Appendix A). Interestingly, the Nrf2-induced transcript *Hmox1* was the fourth most significant DEG and presented a strong upregulation with a log2FC of 3.9 (Figure 4C). We also found upregulation of the additional Nrf2 transcriptional targets *Osgin1*, *Txnrd1*, and *Srxn1*. After 24 h of DMF treatment, we found a lower number of DEGs (264 genes) compared to the vehicle, from which 144 were downregulated and 120 were upregulated (Figure 4B, Appendix A). Most of these DEGs (163) presented strong fold changes, from which around 50% were downregulated and 50% were upregulated (Appendix A). However, we found a higher number of DEGs related to the Nrf2 pathway upregulated at 24 h including *Osgin1*, *Sxrn1*, *Hmox1*, *Gstp1*, *Gsta1*, *Txn1*, *Slc7a11*, *Me1*, and *Gclc* (Figure 4C). At 24 h, the Nrf2 target *Nqo1* was the second most significant DEG with a strong upregulation of log2FC of 2.3. In addition, treatment with DMF for 24 h was the only condition in which we observed a significant upregulation (log2Fc = 0.94) of the gene *Nfe2l2*, which codes for the transcription factor Nrf2. Only nine genes were commonly upregulated between 4 and 24 h after DMF treatment, from which three of them were Nrf2-related (*Osgin1*, *Srxn1*, and *Hmox1*) (Figure 4B,C).

MMF treatment did not produce any significant transcriptomic changes 4 h after drug administration, and it only induced three DEGs after 24 h compared to the vehicle DMSO (Appendix A). Nevertheless, among these three DEGs were *Spta1* and the Nrf2-related *Gstp1* and *Nqo1*, which were both upregulated, although at a lesser magnitude than that with DMF treatment (log2FC of 0.66 and 0.98, respectively) (Figure 4C). CBD modulated only four DEGs (*Slc30a1*, *Mt1*, *Mt1-ps3*, and *Slc30a2*) at 4 h compared to the vehicle, presenting all a strong upregulation with fold changes greater than 1 (Figure 4B, Appendix A, and Appendix A). These genes are related to zinc metabolism, and they have been previously described to be regulated by CBD in microglial cells [40]. On the other hand, after 24 h of CBD treatment, we identified 33 DEGs, 21 downregulated, and 12 upregulated (Figure 4B, Appendix A, and Appendix A). From these DEGs, nine presented a strong upregulation (fold change > 1) and eight presented a strong downregulation (fold change < −1) (Appendix A). In addition to the genes regulated at 4 h, after 24 h since drug treatment, CBD was able to strongly upregulate *Hmox1* (log2FC = 4.9). This upregulation was higher than the one achieved with the 4 or 24 h DMF treatment (Figure 4C). No additional genes related to the Nrf2 pathway were found to be modulated by CBD after 24 h of treatment in neurons.

### 2.5. Transcriptomic Effects of DMF, MMF, and CBD in Microglia

Next, we studied the transcriptomic changes induced by DMF, MMF, and CBD in the LPS-activated microglia cells. First, we examined the effect of LPS treatment compared to the vehicle-treated cells. Samples from the LPS-treated cells separated noticeably in the PCA from the vehicle-only samples at both 4 and 24 h, indicating that LPS exerts strong transcriptomic changes (Appendix A). LPS induced a similar number of DEGs (around 600 genes) when compared to the DMSO- or EtOH-treated samples at both 4 and 24 h (Appendix A). At a given timepoint, most of these DEGs were common, disregarding the vehicle, indicating that LPS and not the vehicle is the main source of the change in gene transcription (Appendix A). At both timepoints, the majority of the DEGs induced by LPS were upregulated (around 400 genes), with downregulation being less intense and with more spread (around 190–250 genes) (Appendix AC–F). As expected, LPS treatment constantly induced the upregulation of genes related to inflammation including activation markers such as *Cd69* and *Cd40*, proinflammatory cytokines like *Il1a*, *Il1b*, and *Tnf,* and chemokines such as *Ccl2*, *Ccl9*, and *Cxcl2* (Figure 5A).

At any given timepoint, independently of the vehicle used, most of the upregulated genes by LPS were common, with similarities ranging from 70 to 90% (Appendix A). Furthermore, when comparing those vehicle-independent commonly upregulated genes at 4 h and 24 h, we still obtained 137 hits (Appendix A). This number still accounted for 44% of the common upregulated genes at any timepoint. These results suggest that, besides the LPS robust expression profile over time, there is also an early and a late induction phase. On the other hand, the downregulation induced by LPS was not shared between the timepoints, indicating an early induction and a late phase (Appendix A). Nevertheless, after 24 h of LPS administration, the downregulation seemed to be highly reproducible between the vehicles as their shared profiles were around 75% similar (Appendix A). All of these results confirm that our murine microglia cell line responds to LPS treatment and is a suitable model to study neuroprotection by DMF, MMF, and CBD.

Next, we checked the gene expression changes in the LPS-treated microglia due to our drugs. Samples treated with DMF formed separated clusters from the LPS-treated samples at both 4 and 24 h in the PCA (Figure 5B,C). In contrast, the MMF samples located near LPS samples at either 4 or at 24 h after treatment. This indicates a strong effect of DMF, and a mild one of MMF. CBD seems to have a delayed effect in microglia transcription. At 4 h, the CBD samples were located close to LPS, in the same fashion as MMF. On the other hand, at 24 h, the CBD-treated samples formed a clearly separated cluster, indicating a transcriptomic effect even stronger than in the DMF-treated samples. When counting the number of DEGs in the drug-treated LPS-induced microglia cells, DMF triggered a total of 542 DEGs (340 down- and 202 upregulated) at 4 h compared to the cells treated with LPS alone (Appendix A). Moreover, from these DEGs, 228 presented a strong downregulation (fold change < −1), and 105 presented a strong upregulation (fold change > 1) (Appendix A). At 24 h, the number of DEGs decreased to 376 (146 downregulated and 230 upregulated) (Appendix A). In contrast to 4 h of DMF treatment, at 24 h, there were more DEGs with a strong upregulation (128 genes) than with a strong downregulation (67 genes) (Appendix A). MMF at 4 h of treatment induced only eight DEGs (three downregulated and five upregulated), which were all common with DMF at the same timepoint (Appendix A). After 24 h of MMF treatment, we could not detect any transcriptomic changes (Appendix A). In the case of CBD treatment, at 4 h, only 2 DEGs were found (one of them common with DMF), but the number of DEGs increased considerably to 2610 (1265 downregulated and 1345 upregulated) at 24 h (Appendix A). Only around 50% of the DEGs presented a strong fold change (>1 or <−1) both in the downregulated and in the upregulated genes (Appendix A) None of the DEGs triggered by 24 h of CBD treatment were common to the ones induced by DMF at the same timepoint.

In order to examine the neuroprotective effect of the drugs, we checked the expression of general inflammatory genes. Only DMF treatment at 4 h showed an important downregulation of the proinflammatory genes activated by LPS, such as the chemokines *Ccl2* and *Ccl7*, the activation markers *Cd40*, *Cd69*, and *Cd83*, and the cytokines *Il1a* and *Tnf* (Figure 5A). However, this downregulation was not maintained at 24 h of DMF treatment. CBD treatment for 24 h also modulated several inflammation genes, but surprisingly, although some were downregulated (*Cxcr2*, *Cd48*, and *Il1b*), we observed that most of the proinflammatory genes were upregulated in the long CBD treatment.

We also studied the regulation by the different drug treatments on genes belonging to pathways of interest (Figure 6). In the case of the NF-kB pathway, LPS produced the upregulation of most of its genes including *Nfkb1*, *Nfkb2*, and *Bcl3* at both 4 and 24 h (Figure 6A). However, none of the drugs produced a relevant modulation of genes from the NF-kB pathway, as only two genes were regulated by DMF (4 h) and CBD (24 h). Notably, the gene *Nfkbid* was commonly downregulated by both treatments. In addition, we observed that the receptor *Hcar2*, one of the proposed targets of DMF, was upregulated by LPS and indeed presented a strong downregulation after 4 h of DMF treatment.

When looking at the Nrf2 pathway, LPS produced no overall effect on it at either 4 or at 24 h (Figure 6B). On the other hand, 4 h of DMF treatment upregulated most of the pathway genes, from which *Hmox1*, *Srxn1*, *Gclm*, *Slc7a11*, and *Osgin1* presented the highest effect. After 24 h of DMF treatment, a lower number of Nrf2-related genes were upregulated, and their induction was also of lower magnitude compared to the early 4 h timepoint. MMF produced the upregulation of *Hmox1*, *Srxn1*, and *Slc7a11* at the 4 h timepoint, but no Nrf2-related genes were found to be modulated at 24 h. CBD showed no regulation of the Nrf2 pathway at 4 h, but at 24 h, there was a general upregulation of several Nrf2-related genes including *Hmox1*, *Srxn1*, and *Gstm1*.

## 3. Discussion

In this study, we examined the potential use of DMF as a neuroprotective agent by analyzing its ability to activate the Nrf2 antioxidant pathway and protect neurons against apoptosis. We also investigated the immunomodulatory effects of DMF in activated microglia by studying the inhibition of the NF-kB pathway and the production of NO. Furthermore, we compared the effects of DMF with its bioactive metabolite, MMF, and the potential use of CBD as an adjuvant therapy in neuroprotection. In addition, we delved into the mechanism of action of DMF, MMF, and CBD by studying the transcriptomic changes induced by the three drugs in neurons and microglia.

In recent years, significant advancements in therapeutics for MS have been achieved due to the approval of numerous therapies with immunomodulatory or immunosuppressant properties. However, the control of neurodegeneration is still an unmet need in MS, highlighting the necessity of developing drugs with demonstrated neuroprotective properties. In this respect, DMF could be relevant beyond its immunomodulatory effects on the peripheral immune system due to its ability to cross the BBB and because it has demonstrated neuroprotective properties in vitro [12,20]. Both DMF and MMF have demonstrated immunomodulatory effects in the EAE model of MS [41]. However, as MMF is the only metabolite detected in plasma, it is considered responsible for the immunomodulatory effects achieved in MS patients [15]. Nevertheless, several studies have suggested that DMF and MMF exert differential effects in vitro in neurons and glial cells, highlighting the necessity of comparing both compounds to evaluate their potential as neuroprotective therapies [26,42,43].

The Nrf2 pathway was initially proposed as the main mechanism of action of DMF [12,44]. However, in Nrf2 deficient mice with EAE, DMF continued exerting clinical benefits and immunomodulatory effects, suggesting that DMF exerts its therapeutic efficacy through other signaling pathways [45]. Nevertheless, activation of the transcription factor Nrf2 could still be relevant for neuroprotection, as the beneficial effects achieved by DMF related to neurodegeneration and oxidative stress in preclinical models appear to be mediated by this pathway [11,46]. Our results confirm, at the protein and transcriptomic level, that DMF is able to activate the Nrf2 pathway in neurons, in line with the published literature, demonstrating the neuroprotective potential of DMF due to the antioxidant properties conferred by this pathway that could protect neurons against the oxidative stress that is produced in inflammatory conditions [9,12]. However, we also found that the biologically active compound MMF did not produce the activation of Nrf2 at the protein level and only produced a slight upregulation of the expression of two Nrf2-related genes at the transcription level, suggesting that, although DMF and MMF are structurally related compounds, they present different abilities to modulate the Nrf2 pathway.

The number of studies comparing the effects of DMF and MMF on the activation of the Nrf2 pathway is scarce, and heterogeneous results have been found between them. One study found that DMF, but not MMF, activated the Nrf2 pathway in the HT-22 mouse and SH-SY5Y human neuronal cell lines [38]. Another study in the SH-SY5Y cell line found a marked upregulation of Nrf2 by DMF, while MMF only produced a partial increase in Nrf2 expression [47]. In the rat oligodendroglial cell line OLN-93, only DMF, but not MMF, increased the protein levels of the downstream target of Nrf2, HO-1 [48]. In primary cultures of human spinal cord astrocytes, both DMF and MMF increased the nuclear levels of Nrf2 [44]. In primary mouse microglia, both DMF and MMF induced transcriptional changes in the Nrf2 pathway, although, it should be noted that DMF produced the activation of a larger number of Nrf2 target genes compared to MMF [42]. Altogether, this suggests that MMF can activate the Nrf2 pathway only in certain cellular types, and that this activation could be more subtle than with DMF treatment. In basal conditions, Nrf2 is located in the cytoplasm due to its sequestration by the Kelch-like ECH-associated protein 1 (KEAP1), which induces the ubiquitination and subsequent proteasomal degradation of Nrf2. Some studies have demonstrated that both DMF and MMF can modify specific cysteine residues on KEAP1 and subsequently activate Nrf2 [12,49,50]. However, different binding affinities to the cysteine residues in KEAP1 have been described between DMF and MMF [50]. In addition, another study showed that protein kinase Cθ has cysteine residues that are only sensitive to DMF and not to MMF [24]. A differential sensitivity of cysteines in KEAP1 to DMF and MMF could be responsible for the differences we observed in Nrf2 activation and should be further investigated. In addition, we observed that CBD also produced the activation of the Nrf2 pathway in neurons and to a higher magnitude than DMF, highlighting its potential as an adjuvant therapy for the achievement of antioxidant properties and neuroprotection. Moreover, we observed that the Nrf2-induced activation by CBD in neurons seemed to be dose-dependent between 2 and 7 µM but showed a decreasing trend in the magnitude of activation beyond 7 µM. A similar effect was found in the HUVEC cells, where CBD induced the activation of Nrf2 at 3 and 6 µM, but not at 10 µM [51]. Similarly, we also observed a slight decrease in Nrf2 activation with DMF 30 µM compared to 10 µM. One possible explanation is that higher doses of the drugs could induce some level of apoptosis, as we observed a decrease in cell viability of around 20% in neurons treated with CBD 10 µM or DMF 30 µM. Previous studies have demonstrated that activation of the proapoptotic protein p53 produces a negative regulation on the Nrf2 pathway to prevent an antioxidant response that could inhibit the induction of apoptosis [52].

Furthermore, in microglia cells, only DMF, but not CBD or MMF, was able to activate Nrf2. The biological relevance of the activation of this pathway in microglia remains unknown, as the immunomodulatory properties achieved by DMF have been demonstrated to be Nrf2-independent [25]. These results also highlight that, although DMF and CBD share mechanisms of action in neurons, they exert different effects in microglia. To the best of our knowledge, there have been no studies evaluating the activation of Nrf2 by CBD in microglia cells, and it would be beneficial to investigate why Nrf2 activation by CBD is limited in microglia in order to understand the mechanism of action of CBD. In this regard, it has been shown that in keratinocytes, CBD can activate the Nrf2 pathway by both increasing the level of activators such as p21 and p62 and by reducing level inhibitors such as Keap1 and Bach1 [53].

In addition to the antioxidant effects attributed to DMF in neurons, it has been suggested that DMF could exert indirect neuroprotective properties through the immunomodulation of microglia and astrocytes [14,26]. Thus, we studied the effect of DMF on the activation of the NF-kB pathway, which is one of the proposed mechanisms of action of DMF [19]. The family of NF-kB transcription factors promotes the upregulation of proinflammatory genes related to different functions including the expression of cytokines and chemokines as well as apoptosis and cell proliferation genes. Microglia cells are the primary immune cells of the central nervous system, and they exhibit activation of the NF-kB pathway in several diseases including MS [54]. NF-kB activation in microglia results in the production of inflammatory mediators that amplify inflammation inside the CNS and lead to demyelination and neurodegeneration, thus constituting a potential therapeutic target in neuroinflammation [55]. In addition, crosstalk between the NF-kB and Nrf2 pathways has been described in several studies, suggesting antagonistic effects between these pathways. NF-kB p65 negatively regulates the transcriptional activity of Nrf2 by competing with the transcriptional co-activator CREB binding protein, resulting in Nrf2 inactivation [56]. On the other hand, the overexpression of Nrf2 inhibits the RAC1-dependent activation of NF-kB p65 and exacerbates inflammation [57].

Our results confirm that DMF can inhibit the activation of the NF-kB pathway by LPS in microglia by preventing the translocation of NF-kB p65 to the nucleus. As with the Nrf2 pathway, we found that MMF did not achieve any effect on the NF-kB pathway, pointing again toward different mechanisms of action of both fumarates. Two previous studies have already described similar results in microglia and in B lymphocytes, where only DMF, but not MMF, inhibited the nuclear translocation of the NF-kB p65 subunit [25,42]. In addition, it has been described that only DMF, but not MMF, modifies the phenotype of activated microglia cells and diminishes the production of proinflammatory cytokines [42,43]. CBD also inhibited the activation of NF-kB by LPS, indicating its potential as an adjuvant immunomodulatory drug in microglia. It is interesting to note that inhibition of the NF-kB pathway by DMF and CBD do not seem to have antagonistic crosstalk with the Nrf2 pathway. Although in our study DMF produced both the activation of Nrf2 and the inhibition of NF-kB, a previous study demonstrated that in the Nrf2 -/- mice, DMF still achieved NF-kB p65 inhibition, thus was Nrf2-independent [25]. CBD inhibited the NF-kB pathway but produced no activation of Nrf2 in microglia, suggesting that the modulation of NF-kB by CBD is also Nrf2-independent.

The inhibition of the NF-kB pathway by DMF and CBD was supported by the inhibition of NO production, which is produced by microglia in proinflammatory states, as its production was reduced by both compounds. Interestingly, we did observe a reduction in the production of NO by MMF, despite the lack of observed effects of MMF on the NF-kB pathway. This reduction in the production of NO by MMF could be explained by the modulation of other pathways such as the HCAR2 pathway, which has been described to be modulated by MMF in microglia [14]. However, the mechanism of action through which MMF could be mediating the inhibition in NO production needs to be further investigated as we did not observe the transcriptomic modulation of *Hcar2* by MMF. We performed transcriptional profiling of the microglia cells treated with MMF to try to shed light on its mechanism of action, however, we could only identify eight differentially expressed genes compared to the LPS-treated cells. From these genes, only CCR5, which is a chemokine receptor that was found to be downregulated, seemed to be related to immune functions. Activation of CCR5 has been linked to the upregulation of iNOS, which synthetizes NO through the activation of NF-κB and secondary pathways via MAPKs ERK, JNK, and p38 [58]. However, we did not observe a modulation of any of these pathways, and the relationship between the downregulation of NO production and CCR5 downregulation by MMF remains to be elucidated. However, the poor transcriptomic response to MMF makes it difficult for us to propose alternative mechanisms of action through which MMF could alter the production of NO.

We also studied the direct potential of DMF, MMF, and CBD in protecting neurons against apoptosis induced by the supernatant of activated microglia cells used as a proinflammatory insult. The three compounds showed the protection of neurons against apoptosis, confirming the potential of DMF as a neuroprotective agent; MMF also presented neuroprotective potential despite presenting differential molecular effects with DMF; and CBD also presented potential as an adjuvant neuroprotective drug in MS. In these experiments, as neurons and microglia were both drug-treated, we could not discern whether the neuroprotection was achieved due to the antioxidant effects in neurons, the immunomodulation of microglia, or both.

Finally, we compared the effect of the three drugs at the transcription level in neurons and microglia. In neurons, DMF exerted the most important transcriptomic changes with a higher number of DEGs and an important upregulation of genes related to the Nrf2 pathway including *Hmox1* and the Nrf2 gene, *Nfe2l2*. In contrast, we did not observe almost any transcriptomic change in neurons due to MMF administration, which, together with the lack of effects observed at the Nrf2 protein level, could indicate a low sensitivity of neurons to MMF. Only a few number of transcriptomic studies have evaluated the effect of fumarates, but it has been described that the transcriptomic changes induced by MMF in astrocytes differ from those achieved by other fumarates such as diroximel fumarate. These findings are in line with the differences found between MMF and DMF in our study [59]. As for CBD, we observed that it achieved only a mild transcriptomic effect in neurons and only modulated one gene from the Nrf2 pathway, *Hmox1*, although it was as highly upregulated as with DMF treatment. Although both DMF and CBD activated Nrf2 at the protein level in neurons, the transcriptomic results suggest that DMF activates the Nrf2 pathway more extensively, modulating a higher number of genes compared to CBD, thus highlighting the potential of DMF in neuroprotection through the activation of antioxidant responses.

In microglia, we observed similar results, with DMF exerting higher transcriptomic changes than MMF, whose effect was weak, highlighting again that the mechanisms of action of both fumarates differ substantially. Regarding CBD, in contrast with the transcriptomic effect observed in neurons, it produced a strong and long-lasting transcriptomic effect in microglia with more than 2500 DEGs, suggesting a higher sensitivity to CBD in microglia cells. DMF produced an overall downregulatory effect in inflammatory genes related to cytokines and chemokines, although the modulation of genes related to the NF-kB pathway was very mild, with only two DEGs, in contrast with the observed inhibition of NF-kB p65 at the protein level. Strikingly, CBD treatment produced an upregulation of cytokine and chemokine genes, although the effect observed on the NF-kB pathway was similar to DMF, with only two DEGs, in contrast again with the inhibition by CBD of the NF-kB p65 protein. Regarding the Nrf2 pathway, DMF and CBD produced the upregulation of a similar number of genes, even though at the protein level, only DMF, but not CBD, achieved the activation of Nrf2 in microglia. These results suggest that both DMF and CBD can activate transcriptomic antioxidant responses in microglia, although their effect on the modulation of inflammatory genes seems less consistent compared to the protein level.

The differential effects found in this study between DMF and MMF highlight the necessity of characterizing the properties of different fumarates for the development of MS therapies. Although the mechanisms of action of both fumarates are not fully understood, it is considered that the therapeutic effects achieved in MS patients are due to MMF, as it is the only metabolite detected in plasma [20]. Furthermore, two MS treatments have been recently approved based on fumarates distinct from DMF. One of them consists only of MMF (Bafiertam^®^, Banner Life Sciences, High Point, NC, USA), and the other consists of a different fumarate named diroximel fumarate (Vumerity^®^, Biogen, Cambridge, MA, USA), which is considered a prodrug of MMF. Although MMF could be responsible for the immunomodulatory effects observed in MS patients, our results suggest that DMF could be more relevant than MMF for the achievement of neuroprotection. This should be taken into consideration for the development of next-generation fumarates that could consist of a combination of MMF and non-metabolizable forms of DMF (conjugated to glutathione) that could achieve summative effects through multiple mechanisms of action. This could be translated into the improvement of clinical benefits, approaching, on the one hand, the immunomodulation of the peripheral immune system, and on the other hand, neuroprotection in the CNS. In addition, our results highlight that CBD has the potential for use as a neuroprotective therapy in MS due to the observed antioxidant and antiapoptotic effects in neurons together with the immunomodulatory benefits in microglia. CBD has demonstrated beneficial therapeutic effects in different murine models of MS including the reduction in inflammatory cell infiltration, axonal damage, and demyelination accompanied with improved clinical signs [30]. In addition, CBD presents a good safety and tolerability profile in MS patients in the form of the approved therapy nabiximol, used for the symptomatic treatment of spasticity [29]. The combination of CBD with other approved MS therapies such as DMF, with which it shares signaling pathways including Nrf2 and NF-kB, could be a relevant strategy to achieve neuroprotection in MS patients. Further studies are needed to evaluate the efficacy and safety of CBD in combination with other therapies.

The combination of DMF, MMF, and CBD could have synergistic therapeutic effects due to their complementary mechanisms of action targeting both oxidative stress and inflammation. Our results suggest that DMF activates the Nrf2 antioxidant pathway more effectively than MMF, promoting the upregulation of antioxidant genes and protecting neurons against oxidative damage. CBD also engages the Nrf2 pathway, with evidence showing that its activation surpasses DMF in magnitude in neuronal cells, further supporting the antioxidant and neuroprotective effects. Additionally, DMF and CBD both inhibit the NF-kB inflammatory pathway, which is central to microglial activation and the production of pro-inflammatory mediators in MS. This dual inhibition could lead to the enhanced suppression of neuroinflammation. MMF, while less active on the Nrf2 and NF-kB pathways compared to DMF and CBD, contributes to anti-inflammatory and neuroprotective effects through distinct mechanisms, which remain to be elucidated. The shared and complementary actions of these compounds (DMF’s strong Nrf2 activation and NF-kB inhibition, MMF’s distinct anti-inflammatory pathways, and CBD’s potent antioxidant and immunomodulatory properties) create a theoretical framework for their synergistic potential in reducing neurodegeneration and inflammation while promoting neuronal survival. This combination could address the multifaceted pathophysiology of neuroinflammatory diseases such as MS more effectively than individual treatments.

While the present study provides significant insights into the individual therapeutic effects of DMF, MMF, and CBD, a limitation lies in the lack of in vitro experiments testing the combination of these compounds. Although the mechanisms of action we describe in this manuscript for DMF/MMF and CBD suggest potential synergistic interactions, direct experimental validation of this synergy should focus on systematically evaluating the combination of the three compounds to better understand their potential interactions. In vitro experiments examining the combined effects of these compounds on pathways such as Nrf2 activation, NF-kB inhibition, and neuronal apoptosis will be crucial. Such studies would provide critical insights into whether their interactions enhance neuroprotection and anti-inflammatory effects, supporting the hypothesis of synergy in vivo. Despite this limitation, the current results provide a strong foundation for the hypothesis that DMF/MMF and CBD may work synergistically. We encourage further investigation to validate and expand upon these findings, which could pave the way for more targeted therapeutic strategies in clinical applications.

## 4. Materials and Methods

### 4.1. Cells

Primary hippocampal neurons were obtained from Wistar rats on embryonic day 18. Briefly, embryos were removed from the womb under sterile conditions and decapitated to dissect the brains. Hippocampi were isolated at 4 °C in Hank’s balanced salt solution (HBSS) without calcium and magnesium, supplemented with 10 mM HEPES buffered saline and 100 U/mL of penicillin-streptomycin. The hippocampi were mechanically disaggregated using fire-polished pipettes and plated for 2–4 h in neuronal plating medium composed of minimum essential medium (MEM) enriched with 0.6% D-glucose and 10% horse serum. Neuronal plating medium was replaced by neurobasal medium supplemented with 1X B-27 and 2 mM L-glutamine. Neurons were grown for 7 days prior to the experiments. For the microglial cells, we used the BV-2 immortalized murine cell line. Microglial cells were cultured in Dulbecco’s modified Eagle medium (DMEM) with 4.5 g/L D-glucose, 2 mM L-glutamine, 100 U/mL of penicillin-streptomycin, and 10% heat-inactivated fetal bovine serum. Cells were maintained at 37 °C in a humidified incubator with 5% CO_2_. All culture media were purchased from Thermo Fisher Scientific (Waltham, MA, USA).

### 4.2. Compound Handling

DMF and MMF (both from Sigma-Aldrich, Saint Louis, MO, USA) were prepared as stock solutions in dimethyl sulfoxide (DMSO); CBD (Cayman chemical, Ann Arbor, MI, USA) was prepared in ethanol (EtOH). Lipopolysaccharide (LPS) from Escherichia coli 026:B6 in an aqueous solution (Thermo Fisher Scientific) was used as proinflammatory stimuli in microglia at a final concentration of 600 ng/mL. For the LPS experiments, the concentration of fetal bovine serum in the microglia culture medium was reduced to 0.1% immediately before drug treatment. All of the compounds were titrated in their respective vehicle and diluted into the culture media for cell treatments. The final concentration of vehicle (DMSO, EtOH or H2O) was consistent for all treated cells (0.2%).

### 4.3. Cell Viability Assay

Neurons and microglia were both seeded at a density of 10,000 cells/well in 96-well plates previously treated with 0.1 mg/mL of poly-L-lysine (Sigma-Aldrich). Cells were treated with different concentration ranges for each drug (1–200 µM for DMF and MMF; 1–30 µM for CBD). Cell viability was determined at 24, 48, and 72 h. The percentage of viable cells was quantified using the CellTiter-Glo^®^ Luminescent Cell Viability Assay (Promega Biotech Ibérica, Madrid, Spain) following the instructions by the manufacturer. The absorbance of the cell culture media alone was subtracted as the background from all readings. Cell viability in the drug-treated cells was normalized to their respective vehicle controls. Experiments were repeated three times, each condition in triplicate.

### 4.4. Immunofluorescence

For the immunofluorescence experiments, neurons and microglia were directly seeded on glass coverslips previously treated with 1 mg/mL of poly-L-lysine. For the Nrf2 experiments, neurons and microglia were treated for 4 h with DMF, MMF, or CBD. For the NF-kB experiments, microglial cells were treated for 30 min with LPS alone or in combination with DMF, MMF, or CBD. After the drug treatments, the cells were fixed on ice with 4% paraformaldehyde for 25 min and then permeabilized and blocked using 0.1% Triton X-100 (10 min) and 5% BSA (30 min), respectively. Cells were then incubated overnight with the primary antibodies and 1% BSA at 4 °C. The primary antibodies anti-Nrf2 (ab1375550) and anti-NF-kB p65 (ab16502) (both from Abcam, Cambridge, United Kingdom) were used at 1:200. After washing with PBS, cells were incubated with the fluorophore-conjugated secondary antibody anti-Rabbit IgG Alexa Fluor 488 (Thermo Fisher Scientific) at 1:500 for 1 h at room temperature. Cells were counterstained with the TO-PRO fluorescent dye (Thermo Fisher Scientific) to visualize the cell nuclei. Images were captured using the confocal microscope Leica TCS SP5 (Leica Microsystems, Wetzlar, Germany). For each experimental condition, five random fields were captured and analyzed using ImageJ software (National Institutes of Health, Maryland, DC, USA). Activation of Nrf2 or NF-kB was defined as nuclear translocation, measured by the increase in the mean fluorescence intensity (MFI) in the nucleus. Each experiment was repeated three times.

### 4.5. Nitric Oxide Production

The concentration of nitrites was measured as an indirect indicator of nitric oxide (NO) production in microglia by using the colorimetric assay Griess Reagent System (Promega Biotech Ibérica, Madrid, Spain). Microglial cells were seeded in 96-well plates, previously treated with 0.1 mg/mL of poly-L-lysine, at a density of 50,000 cells/well. Cells were treated with LPS alone or in combination with DMF, MMF, or CBD for 24, 48, and 72 h. The concentration of NO was quantified at a wavelength of 520 nm in an absorbance reader plate. Experiments were repeated three times, each condition in triplicate.

### 4.6. Apoptosis

The apoptosis of neurons was determined using the DeadEnd™ Fluorometric TUNEL System (Promega Biotech Ibérica, Madrid, Spain) following the manufacturer’s instructions. Neurons were seeded on glass coverslips previously treated with 1 mg/mL of poly-L-lysine at a density of 500,000 cells/well. Neurons were pre-treated for 4 h with DMF, MMF, or CBD. Then, the supernatants of 48 h LPS-treated or LPS + drug (at the same drug concentrations as in neurons) microglia cells were added to the neuronal culture and incubated for an additional 4 h. Apoptosis was determined using Leica TCS SP5 confocal microscopy and quantified using ImageJ software. For each experimental condition, five random fields were captured to count the number of positive fluorescent cells (apoptotic) divided by the total number of cells identified by the nuclear fluorescent staining TO-PRO. Each experiment was repeated three times.

### 4.7. Statistical Analysis

Statistical analysis was performed using Prism 8 software (GraphPad, Boston, MA, USA). For the cell viability assays, a linear regression analysis was made with logarithmic transformation of the drug concentrations in the X-axis. The coefficient of determination (R²) was calculated as a measurement of the adjustment of the data to the regression model. The Mann–Whitney test was used to compare two different conditions. The Kruskal–Wallis test was used to compare the different concentrations of each drug against the vehicle or LPS. Statistical significance was established at *p* < 0.05.

### 4.8. RNA-Sequencing and Bioinformatic Analyses

For the RNA-Seq experiments, neurons were treated with 30 µM DMF, 30 µM MMF, or 6 µM CBD and their respective vehicles for 4 and 24 h. Microglia cells were activated with LPS and treated with the same drug concentrations at the same timepoints. Each experiment was repeated three times. RNA was extracted using the Maxwell^®^ 16 LEV simplyRNA Cells Kit and the robotic platform Maxwell^®^ 16 (both from Promega Biotech Ibérica) following the manufacturer’s instructions. RNA-sequencing was performed as previously described using the NovaSeq 6000 Sequencing System (Illumina, San Diego, CA, USA) by pair-end (150 x 2) and at a sequencing depth of 20 million reads per sample [60]. FastQC v0.12.1 and Cutadapt v4.4 were used for quality control of the raw reads and removal of the adapter sequences, respectively [61]. Trimmomatic v0.39 was employed to further select trimmed reads larger than 20 nucleotides and with a sequencing quality score greater than 30 [62]. Reads were aligned with HISAT2 v2.2.1 to the mouse or rat genome assemblies GRCm39 and mRatBN7.2, respectively. Mapped reads were annotated using Ensembl gene annotation v110 for the mouse genome and v111 for the rat genome and counts quantified using HTSeq v2.0.3 [63]. Gene counts were analyzed using principal components with custom R scripts (R version 4.3.1). DESeq2 v1.40.2 was used for differential expression analysis [64].

## 5. Conclusions

Our study highlights that DMF and MMF present different molecular mechanisms of action, suggesting a higher potential of DMF in neuroprotection due to its ability to activate the Nrf2 antioxidant pathway and inhibit the inflammatory NF-kB pathway. MMF achieved antiapoptotic effects in neurons and anti-inflammatory effects in microglia, but through distinct pathways other than DMF, which should be further investigated. DMF and CBD share common mechanisms of action, with similar antioxidant properties in neurons and anti-inflammatory effects in microglia, highlighting the potential of exploiting CBD as an additional therapy in neuroprotection.

## Figures and Tables

**Figure 1 ijms-25-13082-f001:**
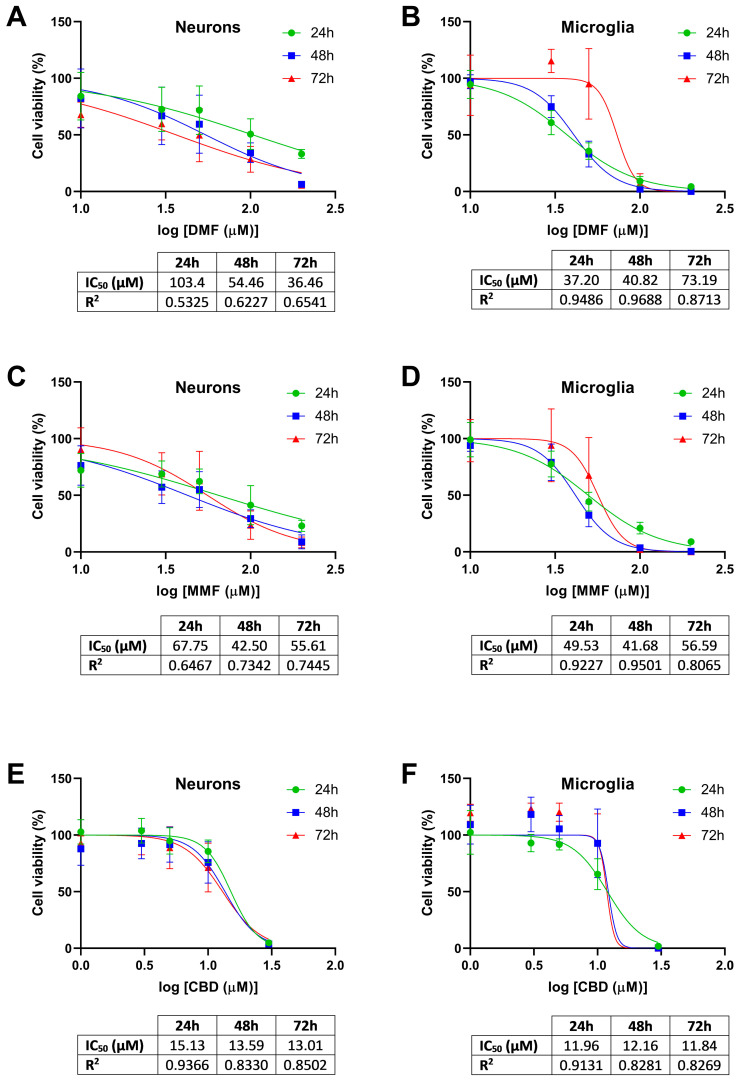
Cell viability of the neurons and microglia cells treated with dimethyl fumarate (DMF), monomethyl fumarate (MMF), or cannabidiol (CBD). (**A**) Cell viability of neurons treated with DMF. (**B**) Cell viability of microglia cells treated with DMF. (**C**) Cell viability of neurons treated with MMF. (**D**) Cell viability of microglia cells treated with MMF. (**E**) Cell viability of neurons treated with CBD. (**F**) Cell viability of microglia cells treated with CBD. (**A**–**F**) Dose–response relationships of each drug at 24, 48, and 72 h. The mean with standard deviation is represented of the percentage of cell viability with respect to the logarithm of the tested concentrations (10, 30, 50, 100, and 200 µM for DMF and MMF; 1, 3, 5, 10, and 30 µM for CBD). The percentage of cell viability was calculated relative to the vehicle-treated cells. The half-maximal inhibitory concentration (IC_50_) and the coefficient of determination (R^2^) values are indicated below each graph.

**Figure 2 ijms-25-13082-f002:**
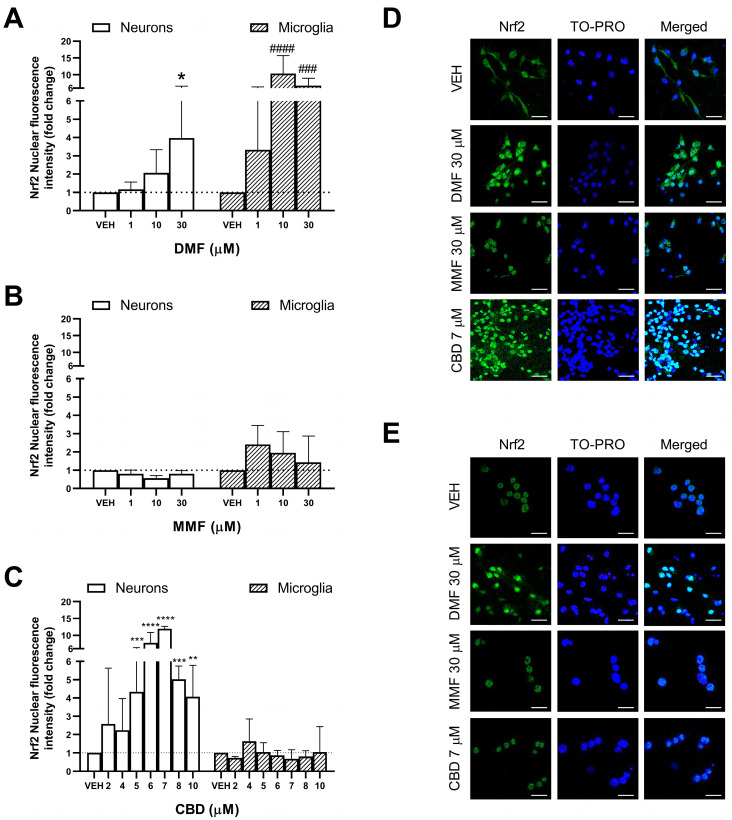
Effect of dimethyl fumarate (DMF), monomethyl fumarate (MMF), and cannabidiol (CBD) on the activation of Nrf2 in neurons and microglia. (**A**) Effect of DMF on Nrf2 activation in neurons and microglia. (**B**) Effect of MMF on Nrf2 activation in neurons and microglia. (**C**) Effect of CBD on Nrf2 activation in neurons and microglia. (**D**) Representative immunofluorescence images from neurons showing the Nrf2 protein location (green). TO-PRO (blue) was used as a nuclear counterstain. Scale bar: 20 µm, shown in the bottom-right corner of each image. (**E**) Representative immunofluorescence images from microglia showing Nrf2 protein location (green). TO-PRO (blue) was used as a nuclear counterstain. Scale bar: 20 µm, shown in the bottom-right corner of each image. (**A**–**C**) White bars represent neurons and black dotted lines represent microglia. (**A**–**E**) Neurons and microglia were treated for 4 h with either the vehicle (VEH) or drug. Bars represent the fold change of the mean nuclear fluorescence intensity of Nrf2 in the drug-treated cells compared to their respective vehicle. Data (mean ± standard deviation) are representative of three different experiments. The Kruskal–Wallis test was used to compare the different concentrations of each drug against their respective vehicle. * *p* < 0.05, ** *p* < 0.01, *** *p* < 0.001, **** *p* < 0.0001 in neurons; ### *p* < 0.001, #### *p* < 0.0001 in microglia.

**Figure 3 ijms-25-13082-f003:**
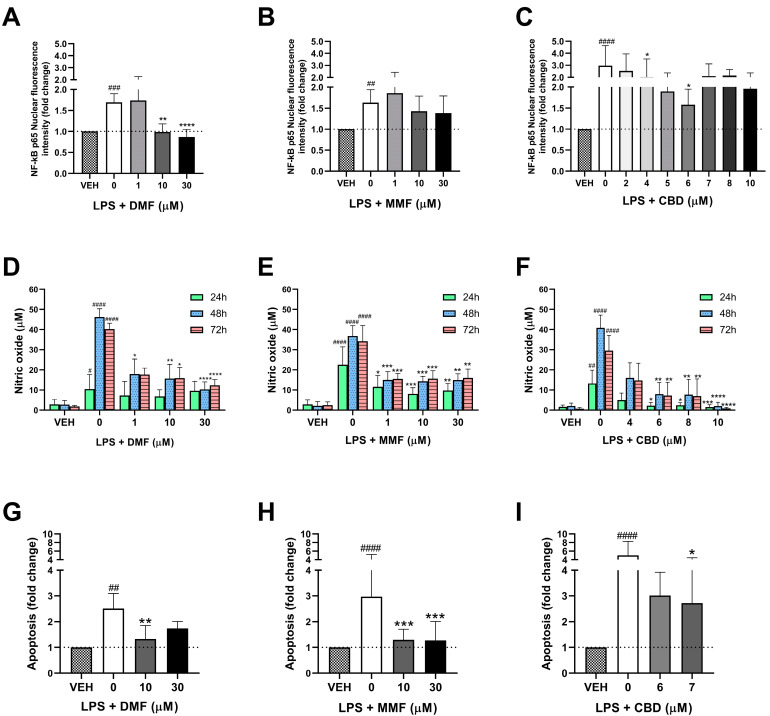
Immunomodulatory effects of dimethyl fumarate (DMF), monomethyl fumarate (MMF), and cannabidiol (CBD) in microglia. (**A**–**C**) Effect of DMF (**A**), MMF (**B**), and CBD (**C**) on the activation of NF-kB p65. Microglia cells were treated for 30 min with either vehicle (VEH), lipopolysaccharide (LPS), or LPS in combination with DMF, MMF, or CBD. Bars represent the fold change of the mean nuclear fluorescence intensity of NF-kB p65 in the drug-treated cells compared to the vehicle (represented by black dotted lines). (**D**–**F**) Effect of DMF (**D**), MMF (**E**), and CBD (**F**) on the production of nitric oxide (NO). Microglia cells were treated for 24, 48, or 72 h with either the VEH, LPS, or LPS in combination with DMF, MMF, or CBD. Bars represent the concentration (µM) of NO. (**G**–**I**) Effect of DMF (**G**), MMF (**H**), and CBD (**I**) on neuronal apoptosis. Neurons were cultured for 4 h with microglia-conditioned medium. Microglia had been previously treated with the VEH, LPS, or LPS in combination with DMF, MMF, or CBD for 48 h. Bars represent the fold change of the number of apoptotic cells (mean ± standard deviation) in the drug-treated cells compared to their respective vehicle. (**A**–**I**) Data are representative of three different experiments. # *p* < 0.05, ## *p* < 0.01, ### *p* < 0.001, #### *p* < 0.0001 LPS compared to the vehicle (Mann–Whitney test); * *p* < 0.05, ** *p* < 0.01, *** *p* < 0.001, **** *p* < 0.0001 LPS + drug compared to LPS (Kruskal–Wallis test).

**Figure 4 ijms-25-13082-f004:**
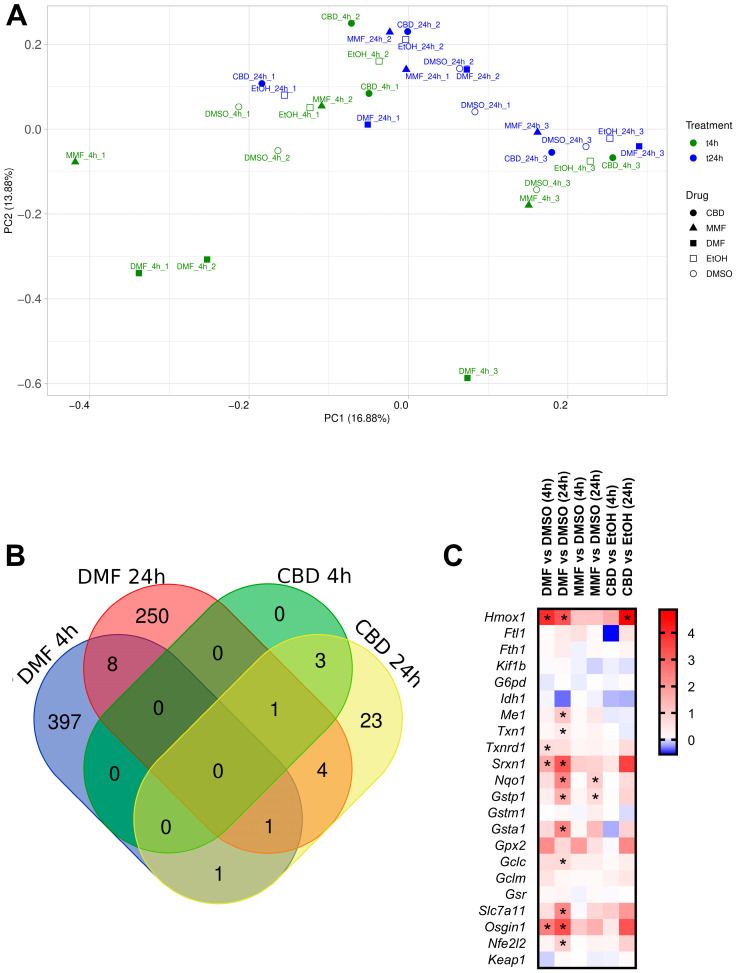
Transcriptomic profile of neurons treated with dimethyl fumarate (DMF), monomethyl fumarate (MMF), or cannabidiol (CBD). (**A**) Principal component analysis of the gene expression profile of neurons treated with 30 µM DMF, 30 µM MMF, 6 µM CBD, or their respective vehicles (DMSO or EtOH) for 4 and 24 h. Each condition presents experimental triplicates. The x-axis represents the first principal component (PC1), and the y-axis represents the second principal component (PC2). The percentages indicated in parentheses represent the percentages of variation explained by the principal components. (**B**) Venn diagram showing the number and overlap of the differentially expressed genes (DEGs) in neurons treated with DMF or CBD for 4 and 24 h. (**C**) Heat map representing the expression levels of genes from the Nrf2 signaling pathway. Gene expression levels are represented with the log2 of the fold change (FC), indicating the ratio of the gene expression in the drug-treated cells compared to their respective vehicle. * *p*-adj < 0.05.

**Figure 5 ijms-25-13082-f005:**
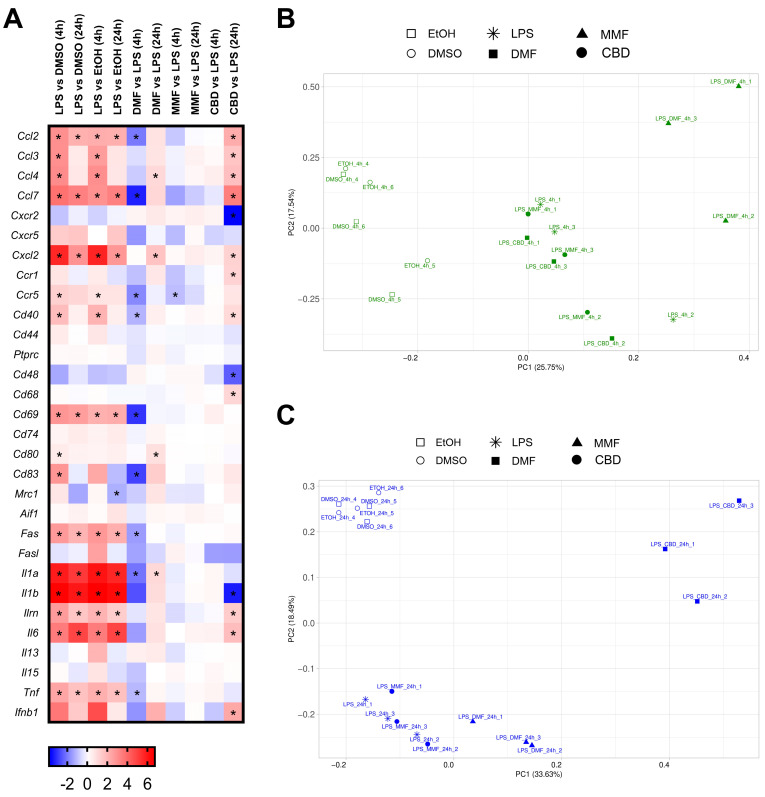
Transcriptomic profile of the lipopolysaccharide (LPS)-activated microglia cells treated with dimethyl fumarate (DMF), monomethyl fumarate (MMF), or cannabidiol (CBD). (**A**) Heat map representing the expression levels of genes related to inflammation. Gene expression levels are represented with the log2 of the fold change (FC), indicating the ratio of the gene expression in the LPS-treated cells compared to the vehicle or drug-treated cells compared to LPS. * *p*-adj < 0.05. (**B**,**C**) Principal component analysis of the gene expression profile of the LPS-activated microglia cells treated with 30 µM DMF, 30 µM MMF, 6 µM CBD, or their respective vehicles (DMSO or EtOH) for 4 h (**B**) and 24 h (**C**). Each condition presents experimental triplicates. The x-axis represents the first principal component (PC1), and the y-axis represents the second principal component (PC2). The percentages indicated in parentheses represent the percentages of variation explained by the principal components.

**Figure 6 ijms-25-13082-f006:**
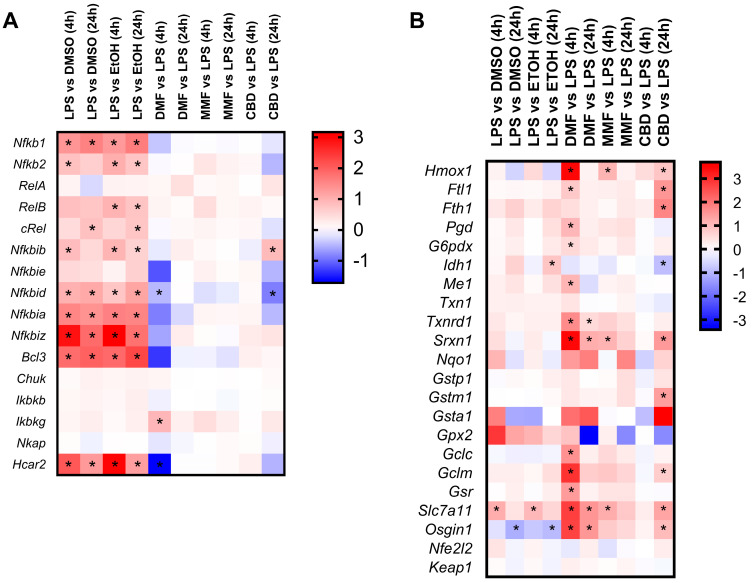
Transcriptomic modulation of the NF-kB and Nrf2 pathways in the lipopolysaccharide (LPS)-activated microglia cells treated with dimethyl fumarate (DMF), monomethyl fumarate (MMF), or cannabidiol (CBD). (**A**) Heat map representing the expression levels of genes related to the NF-kB pathway. (**B**) Heat map representing the expression levels of genes related to the Nrf2 pathway. (**A**,**B**) LPS-activated microglia cells were treated with 30 µM DMF, 30 µM MMF, 6 µM CBD, or their respective vehicles (DMSO or EtOH) for 4 and 24 h. The gene expression levels are represented with the log2 of the fold change (FC), indicating the ratio of the gene expression in the LPS-treated cells compared to the vehicle or drug-treated cells compared to LPS. * *p*-adj < 0.05.

## Data Availability

The raw RNA-sequencing data of the microglia cells and neurons have been deposited in the NCBI’s public repository Gene Expression Omnibus (GEO) and are accessible through the accession numbers GSE274859 and GSE274435, respectively.

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
