# Peer review of "Neuroprotective and Anti-Inflammatory Effects of Dimethyl Fumarate, Monomethyl Fumarate, and Cannabidiol in Neurons and Microglia"

_ijms, 2024, doi:10.3390/ijms252313082_

Round 1
Reviewer 1 Report
Comments and Suggestions for Authors
Neuroprotective and anti-inflammatory effects of dimethyl fumarate, monomethyl fumarate and cannabidiol in neurons and microglia
In this paper, the researchers investigated the neuroprotective effect of DMF, its bioactive metabolite MMF and adjuvant CBD in microglia. They found that DMF and MMF had different properties, and that CBD has also potential in neuroprotection. They first investigated the IC50 of each compounds in neuron extracted from rats and immortalized murine cell lines. After, they investigated Nfr2 activation at concentration below the IC50, and showed that DMF were effective in increasing Nfr2 expression while CBD only in neuron, and MMF failed at statistical significant level using IHC. They also used IHC combined with cell viability assay kit to evaluate the immunomodulatory effects of DMF, MMF and CBD. They also evaluated the effects of these compounds in inflammatory pathways using transcriptomics in neuron and microglia.
Major comments:
1. There is a question that are needed to be addressed. As MMF is a bioactive metabolite of MMF and CBD is an adjuvant, it will be interesting to observe the effect of DMF and MMF or all three of these compounds to assay if any synergistic or inhibitory effects are observed.
2. It is necessary to clearly explain the connection between the analysis of NRF2 changes in Result 2.2 and the analysis of anti-inflammatory effects in Result 2.3. An additional explanation is needed on how changes in NRF2 influence anti-inflammatory effects or how these two findings are related.
3. The discussion needs to provide a more detailed explanation of the differences by which transcriptome DMF, MMF and CBD differ in their neuroprotective mechanisms.
4. Authors state that when DMF is administered orally, it is hydrolyzed to MMF, mostly by intestinal esterase, so how did you evaluate and confirm that DMF passes the BBB? Please provide further explanation.
5. In the introduction, authors claimed that there are metabolic and pharmacological differences between DMF and MMF, what exactly are these differences? Please add more explanation. (How the sensitivity of the two fumarates is different and what it means, what are the different effects on the NF-kB pathway, GSH responsiveness, glial cell regulation).
6. One of the aims of the study was to evaluate the possibility of CBD being used as an adjunctive treatment to DMF, which has neuroprotective and anti-inflammatory properties. In order to increase the validity of the paper, shouldn't the study also include a co-treatment of DMF and CBD, in addition to the treatment with DMF and CBD individually, to see if a synergistic effect might occur?
7. In Figure 2, DMF effectively activates the NRF2 antioxidant pathway, but MMF appears to have a minimal effect. Further clarification or experiments are needed to understand the mechanism by which DMF contributes to NRF2 pathway activation. Additionally, the effects of CBD on NRF2 activation differ between neurons and microglia. It would be beneficial to investigate why NRF2 activation is limited in microglia.
8. While CBD shows promise as an adjunct therapy, studies are needed to assess its long-term efficacy and safety when used in combination treatments.
Minor comments:
1. The sentence of line 29, “Transcriptomically, DMF, MMF and CBD exhibited differential effects on these pathways, with DMF achieving the most pronounced changes.”, should be a little more specific.
2. It would be better to add more relevant reference explaining the effect of the NF-kB pathway on the neuroprotection.
3. In figure 2D, it would be better to also display confocal microscopy images of microglia.
4. Also, Immunomodulatory effects of DMF, MMF and CBD in neuron need to be investigated.
5. In Figure 1, both IC50 and R2 values are shown with a decimal point of , instead of . It would also look good to add statistical significance. Also, the x-axis values for E and F started at 0.0, why do A, B, C, and D start at 1.0?
6. In Figure 2, why is there no immunofluorescence experiment confirming Nrf2 translocation in microglia?
7. In Figure 3, shouldn't the NF-kB activation, the amount of NO, and the percentage of apoptotic cells that occur when treated with LPS alone (the 0 part of the x-axis) be almost similar or the same in the three cases (DMF, MMF, CBD)? Why is there a difference in the amount of NO produced when the same microglia cells were treated with LPS without any substance treatment?
8. In analyzing PCA and DEG, what is the reason for the division by 4h and 24?9. What do the x-axis and y-axis (PC1, PC2, respectively) in the figure representing the results of the PCA analysis mean? This is not explained.
9. It is not clear from the figure what log2FC is, which is mentioned in the text when describing Figure 4C. Please include in the figure.
Reviewer 2 Report
Comments and Suggestions for Authors
In this manuscript, Sanchez-Sanz et al. investigate the Neuroprotective and anti-inflammatory effects of Dimethyl Fumarate, Monomethyl Fumarate and Cannabidiol in neurons and microglia. The authors explored the roles of these compounds by performing various in vitro approaches using primary neurons from wistar rats along with murine microglial cells and presented an analysis of the data accordingly. A wide variety of methods were performed to corroborate their hypothesis. However, I would like to request the authors to address the comments below regarding this manuscript.
1. The authors have not stimulated the cells for the activation of Nrf2 pathway and have treated the cells with DMF,MMF and CBD to show their effects on Nrf2 translocation. Can the authors provide some clarification about why this experiment was performed with out any activating stimuli ?
2. CBD has a dose dependent effect on the Nrf2 activation in the neurons but tend to show a decreasing trend beyond 7uM concentration range. It would be helpful if authors can explain this in the discussion/results section.
3.DMF have also showed a dose dependent effect in Nrf2 activation in microglial cells till 10uM range but have proceed to show a decreasing trend at 30uM. It would also be helpful if the authors can explain this further.
4. The authors haven't included any Immunofluorescence images for the NF-kB nuclear translocation experiment but have presented the quantified (fold change) data for the same. If the authors can provide this data set, it would be helpful.
5. NO secretion was measured as an indicator of an proinflammatory state in LPS stimulated microglia. The authors have demonstrated MMF had significantly inhibited NO secretion at 24hr compared to DMF. It would be helpful if authors can explain if they have looked into any other inflammatory markers apart from NO and experienced the same phenomenon.?
6. The authors haven't provided any con-focal microscopic images for the apoptosis assay (Fig 3G-I).
7.Have the authors looked into any Demyelination markers to show the effects of CBD ?
8. It would be helpful if the authors can summarize their results in the form a pictorial abstract to represent the potential targets of DMF MMF and CBD.
Round 2
Reviewer 1 Report
Comments and Suggestions for Authors
Neuroprotective and anti-inflammatory effects of dimethyl fumarate, monomethyl fumarate and cannabidiol in neurons and microglia
This manuscript is suitable for submission to the journal except comment 1. In the comment 1 response, the authors suggest that DMF and MMF combination alongside CDB adjuvent will have synergistic therapeutic effects. However, effect of those chemical in cells were only assayed individually. It is understandable that as MMF is a metabolite its effect in vivo systems will be assayed by DMF treatment. However, in vitro experiments of the effect of DMF, MMF and CDM combination can be done to prove that combination will be synergestic in vivo.
Reviewer 2 Report
Comments and Suggestions for Authors
I would like to thank the authors for making the revisions and providing the updated manuscript. After going through the revisions, I would like to add a few more corrections that needs to be revised
1. The supplementary figure for apoptosis demonstrates that the authors had obtained images at a different settings/exposure for the CBD group compared to the other groups.. Can the authors please explain this ?
2. The relation between MMF and CCR5 needs to be mentioned in the discussion section to provide some clarity
3. It would be helpful if the authors can include the scale bar in their Immunofluorescence images.
4. Description for fig 2 is missing in lines 797.
